# Development of Novel Microwell-Based Spectrofluorimetry and High-Performance Liquid Chromatography with Fluorescence Detection Methods and High Throughput for Quantitation of Alectinib in Bulk Powder and Urine Samples

**DOI:** 10.3390/medicina59030441

**Published:** 2023-02-23

**Authors:** Halah S. Almutairi, Mohammed M. Alanazi, Ibrahim A. Darwish, Ahmed H. Bakheit, Mona M. Alshehri, Hany W. Darwish

**Affiliations:** Department of Pharmaceutical Chemistry, College of Pharmacy, King Saud University, P.O. Box 2457, Riyadh 11451, Saudi Arabia

**Keywords:** alectinib, 96-microwell spectrofluorimetry, HPLC, high throughput analysis, urine samples

## Abstract

*Background and Objectives:* This study presents the development and validation of the 96-microwell-based spectrofluorimetric (MW-SFL) and high performance liquid chromatography (HPLC) with fluorescence detection (HPLC-FD) methods for the quantitation of alectinib (ALC) in its bulk powder form and in urine samples. *Materials and Methods:* The MW-SFL was based on the enhancement of the native fluorescence of ALC by the formation of micelles with the surfactant cremophor RH 40 (Cr RH 40) in aqueous media. The MW-SFL was executed in a 96-microwell plate and the relative fluorescence intensity (RFI) was recorded by utilizing a fluorescence plate reader at 450 nm after excitation at 280 nm. The HPLC-FD involved the chromatographic separation of ALC and ponatinib (PTB), as an internal standard (IS), on a C_18_ column and a mobile phase composed of methanol:potassium dihydrogen phosphate pH 7 (80:20, *v*/*v*) at a flow rate of 2 mL min^–1^. The eluted ALC and PTB were detected by utilizing a fluorescence detector set at 365 nm for excitation and 450 nm for emission. *Results:* Validation of the MW-SFL and HPLC-FD analytical methods was carried out in accordance with the recommendations issued by the International Council for Harmonization (ICH) for the process of validating analytical procedures. Both methods were efficaciously applied for ALC quantitation in its bulk form as well as in spiked urine; the mean recovery values were ≥86.90 and 95.45% for the MW-SFL and HPLC-FD methods, respectively. *Conclusions:* Both methodologies are valuable for routine use in quality control (QC) laboratories for determination of ALC in pure powder form and in human urine samples.

## 1. Introduction

Alectinib (ALC) is a small molecule (Figure 1) chemically named as 9-ethyl-6,6-dimethyl-8-[4-(morpholin-4-yl)piperidin-1-yl]-11-oxo-6,11-dihydro-5H-benzo[b]carbazole-3-carbonitrile HCl [1]. ALC is an effective second-generation anaplastic lymphoma kinase (ALK) inhibitor, shown to be effective for a broad spectrum of ALK mutations [2]. The Food and Drug Administration (FDA) has granted fast approval to the drug ALC for treating people suffering from ALK-positive non-small-cell lung cancer (NSCLC) and experiencing resistance to crizotinib, or if their disease has progressed despite therapy with crizotinib [3]. This accelerated approval was supported by the robust efficacy and safety of ALC in clinical phase I/II pivotal studies. ALC has also proved to be beneficial to patients with brain metastases [4,5,6]. In addition, ALC has shown a positive effect on the central nervous system (CNS) penetration and safety profile [7,8,9,10].

Hepatotoxicity and nephrotoxicity commonly occur in patients receiving ALC [11,12]; therefore, a thorough understanding of the interactions between the medication level and its activity is critical for its safe and effective use. Therapeutic drug monitoring (TDM) mainly depends on the accurate and precise measurement of the drug level in the biological samples. Patients’ clinical assessments and choices around additional treatment alternatives are determined by the results of a validated trustable analytical method [10,13]. Therefore, an efficient and reliable analytical technique is required for quantitation of ALC.

The existing techniques for the quantitation of ALC are very limited and they mainly involve high-performance liquid chromatography coupled with an ultraviolet detector (HPLC-UV) [14], HPLC with a photodiode array detector (HPLC–PDA) [15], and liquid chromatography with tandem mass spectrometry (LC-MS/MS) [15,16,17,18,19,20,21]. LC-MS/MS displays excellent selectivity and sensitivity; however, it is expensive and demands specialized technical expertise, which might not be easily accessible or within the budget range of the majority of laboratories in nations with low resources. The existing HPLC-PDA has adequate selectivity; however, it lacks high sensitivity. In addition, these methods have a limited throughput for routine application to the analysis of bulk samples for either quality control or in clinical laboratories. This work describes the development and validation of two new methods with high sensitivity, simple procedures, and a high throughput for ALC quantitation in its bulk form and for urine samples. Microwell-based spectrofluorimetric (MW-SFL) and HPLC with fluorescence detection are these two methodologies (HPLC-FD).

## 2. Experimental

### 2.1. Instruments

A cuvette-microplate multifunction (absorbance, fluorescence and chemiluminescence) reader spectramax M5 (Molecular Devices, San Jose, CA, USA) was used for spectrofluorometric measurements. The data acquisition was achieved utilizing Softmax Pro^®^GXP software provided with the instrument. The HPLC system (Waters, Milford, MA, USA) consisted of a Waters 1525 binary HPLC pump, a Waters 2475 fluorescence detector, and a Waters 2707 autosampler. The detector was set at 365 and 450 nm for excitation and emission, respectively. The data were acquired and processed using Windows XP-based Waters Breeze 2 software. IKA was contacted for the acquisition of a magnetic stirrer (Wilmington, NC, USA). Sonicator cleaning technology that uses ultrasonic waves (X-TRA150H, Elma, Bedford, UK) was utilized through the study. A digital electric balance was procured from Mettler-Toledo International Inc. (Zurich, Switzerland). A Milli-Q^®^ water purification system (Millipore Ltd., Bedford, MA, USA) was used.

### 2.2. Chemicals and Reagents

Alectinib (ALC) standard was purchased from Med. Chem. Express (Princeton, New Jersey, NJ, USA). Ponatinib (PTB) was purchased from Sigma-Aldrich Corporation (St. Louis, MO, USA). White opaque 96-microwell plates were a product of Corning/Costar Inc. (Cambridge, MA, USA). Finnpipette adjustable 8-channel pipette was obtained from Sigma Chemical Corporation (St. Louis, MO, USA). Analytical grade surfactants PEG-40 Hydrogenated Castor Oil with ethylene oxide (cremophor RH 40: Cr RH 40) and Polyethoxylated castor oil (cremophor EL: Cr EL) were purchased from BASF (Ludwigshafen, Germany). Sodium dodecyl sulfate (SDS) was obtained from WINLAB (Pontefract, London, UK). Carboxymethyl cellulose (CMC) was purchased from Merck (Darmstadt, Germany). Polyoxyethylene (20) sorbitan monooleate (Tween 20) and polyoxyethylene (80) sorbitan monooleate (tween 80) were obtained from Techno Pharmchem Haryana Company (New Delhi, India). All surfactants were as 1% solution in water. The freezing points of these surfactants were less than −20 °C. A reverse-phase Eclipse plus C_18_ HPLC column (5 µm, 4.6 mm i.d × 250 mm) was a product of Waters Corporation (Milford, MA, USA). Each solvent was of a spectroscopic or chromatographic grade (Merck, Darmstadt, Germany). All additional chemicals utilized for the project’s entire duration were of an analytical grade. Healthy adult male participants provided urine samples that were stored frozen until analysis.

### 2.3. Preparation of Standard Solutions

The ALC standard powder (12.5 mg) was transferred to a 25-mL volumetric flask and dissolved in dimethyl sulfoxide (DMSO) to achieve a desired final concentration of 0.5 mg mL^−1^. If refrigerated, this stock solution would keep for at least 2 months. A working standard solution with a final concentration of 500 ng mL^−1^ was obtained by transferring 250 µL of this stock solution into a 25-mL volumetric flask and brought up the volume to the mark with acetonitrile.

### 2.4. Preparation of Urine Samples

Human urine samples of 1 mL were spiked with 20 μL of ALC standard solution containing different amounts of ALC, and the contents were vortexed for 30 s (using a stop-watch). A 1 mL buffer of NaOH 100 mM/glycine (pH 11) was added to this solution, and the contents were stirred for 10 s. These samples underwent liquid–liquid extraction with 5 mL of diethyl ether, followed by 15 min of centrifugation at 10,000 rpm for phase separation. The upper organic layer was collected in a glass vial and dried over a moderate stream of nitrogen. Residues were reconstituted in acetonitrile and diluted to final ALC concentrations of 60, 120, 240, and 480 ng mL^–1^. The ALC content of these samples was evaluated using both the MW-SFL and HPLC-FD methodologies.

### 2.5. General Analytical Procedure

#### 2.5.1. MW-SFL

Precisely measured aliquots (100 µL) of the standard or sample (either bulk form or urine) solution containing 30–500 ng mL^−1^ of ALC were relocated into wells of the microwell plates. A 60 µL of Cr RH 40 solution (1%, *w*/*v*) and 40 µL of phosphate buffer (pH 7) were added sequentially. The relative fluorescence intensity (RFI) of all the prepared samples was recorded utilizing a fluorescence reader at 450 nm for emission (after excitation at 365 nm).

#### 2.5.2. HPLC-FD

On a Reverse-phase Eclipse plus C18 HPLC column (5 m, 4.6 mm i.d. 250 mm), ALC was analyzed by HPLC where temperature of the column was maintained at 25 ± 2 °C. ALC and PTB (IS) were separated using the isocratic mode for elution with a mobile phase of methanol:potassium dihydrogen phosphate of pH 7 (80:20, *v*/*v*) utilizing a 2 mL min^−1^ flow rate. The mobile phase was degassed via 15 min of helium gas pumping and a 10-min ultrasonic bath. The volume of the sample injection was 10 μL. The excitation wavelength of the fluorescence detector was set to 365 nm and the emission wavelength was set at 450 nm. For the production of the ALC calibration curve, standard calibration solutions containing different concentrations of ALC (5–1000 ng mL^−1^) and a constant concentration of PTB (15.6 ng mL^−1^) were prepared. The quantification was based on the relationship between the peak area ratios of analyte (ALC) to IS (PTB) on the *Y*-axis and the analyte concentration (*X*-axis). ALC was calculated from its sample data using the obtained regression equations.

## 3. Results and Discussion

### 3.1. Development of the MW-SFL Method

#### The Strategy for Method Development and Its Design

The investigation of pharmaceutical and biological substances frequently makes use of a technique called spectrofluorimetry. There are numerous uses for spectrofluorimetry as a result of its intrinsic ease of use, high sensitivity, and relatively inexpensive analysis [22,23,24]. A preliminary investigation demonstrated that ALC had a native fluorescence, as was hypothesized based on its chemical structure (Figure 1). This is due to the presence of fluorophoric moieties inside the molecule, such as prolonged conjugations and rigid fused rings. In order to construct the method discussed in this research, spectrofluorimetry was explored. The conventional spectrofluorimetric procedures have a throughput that is significantly lower than that of their automated counterparts because they are non-automated the majority of the time [22,23]. In addition, enormous amounts of expensive organic solvents are used in these procedures, and, perhaps more crucially, the analyst is put in direct contact with the hazardous byproducts of the organic solvents [24]. Recent developments in our laboratory have resulted in the successful adaptation of a fluorescence plate reader for use in the creation of a microwell-based spectrofluorometric approach to assess the pharmaceutically active content in dosage form [25]. This technique has a high throughput and requires only a small amount of the respective organic solvents and samples. The construction of this methodology for ALC considered this information.

It has been shown that the sensitivity for the quantification of small drug molecules can be improved by using micelle-enhanced spectrofluorimetry [26,27]. Micelle formation is crucial because it elevates the RFI of pharmaceuticals with low native fluorescence. The absence of an organic solvent makes micelle-enhanced spectrofluorimetry a green chemistry method that is both efficient and effective. These methods depend on using a surfactant to enhance the fluorescence signal of the drug molecule and ultimately the sensitivity of the spectrofluorimetric method [28]. A previous study [29] demonstrated the successful use of Cr RH 40 as an excellent fluorescence-enhancer surfactant in constructing micelle-based spectrofluorimetric methodology for quantitation of drugs in their dosage forms and in biological samples. For these reasons, the current work was undertaken to develop a microwell-based micelle-enhanced spectrofluorometric method for ALC.

### 3.2. Spectral Featutres

The excitation and emission fluorescence spectra of a methanolic solution of ALC were recorded in the absence and presence of Cr RH 40 (Figure 2). The spectra of ALC showed that ALC has three excitation wavelengths (λ_ex_) at 260, 275, and 365 nm and showed a maximum emission (λ_em_) at 450 nm. Furthermore, the existence of Cr RH 40 gave rise to the marked enhancement of the ALC fluorescence intensity, when compared with its intrinsic intensity in aqueous media. The primarily realized intensity of the enhanced fluorescence of ALC was promising for developing a sensitive spectrofluorometric method for its quantitation in both bulk form and urine samples. In all the following experiments, the responses were measured at 450 nm (after excitation at 365 nm).

### 3.3. The Optimization of the MW-SFL Method Variables

#### 3.3.1. The effect of pH

The influence of pH on the RFI of ALC was tested, utilizing different buffers in the pH range of pH 3–11. The results revealed the dependence of the RFI values on the pH of the medium and the maximum RFI value was obtained at pH 7 and 11, respectively, (Figure 3), and all the subsequent experiments were carried out at pH 7.

#### 3.3.2. The Effects of Types of Surfactant

The effects of the types of surfactant on the RFI values of ALC was tested by the addition of 40 µL of each particular surfactant’s aqueous solution (1%, *w*/*v*) to each well containing the ALC solution (100 µL). These surfactants were categorized into non-ionic (Cremophor RH 40 (Cr RH 40), Cremophor EL (Cr EL), tween 20, and tween 80), anionic (SDS), and macromolecules (CMCs). The highest RFI was attained by utilizing the non-ionic Cr RH 40 and Cr El (Figure 4). This was attributed to the fact that the non-ionic surfactants have a higher solubilization power for hydrophobic drugs (ALC), compared with the ionic surfactants [30].

#### 3.3.3. The effect of Cr RH 40 Volume

The impact of Cr RH 40 volume on the RFI was tested using different volumes (15–70 µL) of Cr RH 40 solution (1%, *w*/*v*). The results revealed that the RFI increased as the volumes of Cr RH 40 solution increased, and the maximum value of RFI was obtained at 50 µL (Figure 5). Beyond this volume, no further increase in RFI was observed. For reading with better precision, 60 µL of Cr RH 40 solution (1%, *w*/*v*) was utilized in the later experiments.

#### 3.3.4. The Effect of the Solvent

Various solvents were tested to assess their effects on the RFI of ALC-Cr RH 40 medium. These solvents were water, acetonitrile, methanol, and ethanol. It was found that water gave the highest RFI values, compared with the other solvents (Figure 6). This may be attributed to the high polarity effect of water, which promotes the physical interaction between Cr RH 40 and the excited singlet state of ALC. Accordingly, water was used as a solvent throughout the following experiments. The decrease of the RFI of ALC in a Cr RH 40 medium, in the existence of methanol, ethanol, and acetonitrile, could be due to these solvents’ abilities to denature the micelles. Methanol and ethanol are predominantly dissolved in water, which is similar to the short-chain alcohols. They alter the characteristics of the solvents, which affects the creation of the micelle. Additionally, these short chain alcohols might decrease the micellar size and might decompose the surface-active agent aggregate at high concentrations [30,31].

Table 1 provides a description of the variables that were evaluated across a range of values and shows which value was chosen as the optimal one for the construction of the MW-SFL for ALC.

### 3.4. The Mechanism of the Micellar Enhancement of ALC Fluorescence by Cr RH 40

The fluorescence enhancement of ALC might be due to either the rise in the quantum yield and/or an enhanced absorption at the excitation wavelength (365 nm). A molar absorptivity (ε) evaluation of ALC in a Cr RH 40 medium was accomplished at 365 nm. The ε micellar/ε water ratio was 2. The ALC quantum yield was 0.0217 in water and 0.1272 in the micellar Cr RH 40 medium. Therefore, the enhancement of the RFI is due to enhanced absorption at λ_ex_ and an increase in quantum yield. This rise in the ALC quantum yield in the presence of Cr RH 40 may be an effect of the lowest excited singlet state protection against non-radiative processes. Computation of quantum yield was applied according to the following equation:$$\Phi_{F}=\Phi_{F\left(\mathrm{Std}\right)}\;.\;\frac{F\;.\;A_{\mathrm{Std}}\;.\;n^{2}}{F_{\mathrm{Std}.}\;.\;A\;.\;n_{{\mathrm{Std}}^{2}}}$$
where Φ_F_ and Φ_F(Std)_ represent the quantum yields of ALC and quinine, respectively. F and F_Std_ represent the fluorescence intensities of ALC and quinine; A and A_Std_ are the absorbance values of ALC and quinine at the ex, where n and n_Std_ are the refraction indexes. The concentration of ALC that had an absorbance of less than 0.05 was selected, to diminish the error that might arise from the inner effect [32].

### 3.5. The Development of the HPLC-FD Method

#### 3.5.1. The Overview and Method Development Strategy

The widespread use of HPLC with fluorescence detection (HPLC-FD) in pharmaceutical and biomedical analyses is a result of its innate superior sensitivity and convenience. In addition, it can be modified as a high throughput method if it is equipped with an autosampler and has a brief run time. Extensive research has found that there is no HPLC-FD method for quantifying ALC. Since ALC, as confirmed by its fluorescence spectra (Figure 2), has a native fluorescence, the present study was devoted to the development of an HPLC-FD method with a high throughput for its determination in bulk and/or urine samples.

#### 3.5.2. The Selection of Chromatographic Conditions

##### The Selection of Detection Wavelength and Internal Standard

The fluorescence spectra of ALC (Figure 2) indicated that ALC has its maximum excitation wavelength (λ_ex_) at 365 and its maximum emission wavelength (λ_em_) at 450 nm. These excitation and emission wavelengths were selected for detection of ALC. For selection of a proper internal standard (IS), different molecules of the tyrosine kinase inhibitors’ family were tested. These molecules were ponatinib (PTB), erlotinib, dasatinib, and vandetanib. The criteria used in the selection of the proper IS were having a good separation under the chromatographic conditions used for ALC and the detectability of ALC at the wavelengths specified. It was found that PTB meets these criteria, and accordingly, it was selected as an IS.

##### Mobile Phase

In order to determine the optimal mobile phase, we conducted a series of experiments to determine its components and flow rate. An isocratic elution mode was used to start the chromatographic separation on a C_18_ HPLC column (5 m, 4.6 mm i.d., 250 mm). The column was maintained at a constant temperature of 25 ± 2 °C. The mobile phase for the separation experiments consisted of various ratios of phosphate buffer solution with either methanol or acetonitrile. The results revealed that a mixture of phosphate buffer–methanol was better than phosphate buffer–acetonitrile as ALC was eluted within 10 min with a sharp peak. Therefore, methanol was selected as the proper organic modifier in the subsequent experiments. Varying ratios of phosphate buffer:methanol (60–90%, *v*/*v*) were tested. At a lower methanol percentage, forking and tailing of the ALC peak occurred, along with a prolonged retention time. At higher methanol percentages, sharper peaks were obtained at shorter retention times. The best resolution with the sharpest peak was achieved when the ratio of phosphate buffer:methanol was 20:80 (%, *v*/*v*). The pH of phosphate buffer solution was tested to establish whether it could influence the charge state of the ionizable species in solution. The extent of analyte ionization can be used to affect retention and selectivity. The optimum pH of the buffer solution could control the elution properties by controlling the ionization characteristics of the drug. It can also decrease the retention time and improved the resolution. Different pH values ranging from 4–7 were tested; the best was pH 7. Different flow rates (0.5–2 mL min^–1^) were studied with reference to the column back pressure. The optimum flow rate was 2 mL min^–1^, because it gave the best resolution within a short retention time. Under these best chromatographic conditions, ALC and PTB were eluted as sharp narrow symmetrical peaks at 4.86 and 5.92 min, respectively (Figure 7).

##### Suitability Parameters

System suitability parameters are utilized to confirm the adequacy of the system reproducibility and resolution. The parameters (retention time, capacity factor, separation factor, resolution factor, peak asymmetry factor, and the theoretical number of plates) were computed in accordance with International Council for Harmonization (ICH) guidelines [33]. The obtained results (Table 2) were found in accordance with and demonstrating good performance efficiency. The chromatographic characteristics show that the proposed HPLC-FD method allowed an excellent resolution within an acceptable run-time (appropriate capacity factors). High column efficiency was designated from many theoretical plates. It was also determined, with the help of the tailing factor, that the extent of the peak asymmetry did not go over the crucial value (1.14). This is an indication of acceptable peak asymmetry.

### 3.6. The Validation of the MW-SFL and HPLC-FD Methods

#### 3.6.1. Linearity and Sensitivity

Calibration curves were built under the predetermined revised optimal conditions for the MW-SFL and HPLC-FD methodologies (Figure 8), and the least-squares approach was used to analyze the data using linear regression. For the MW-SFL and HPLC-FD the graphs were linear with remarkable correlation coefficients’ values in the ranges of 30–500 and 5–1000 ng mL^−1^, respectively. Table 3 lists different parameters for calibration of the MW-SFL and HPLC-FD methods.

According to the ICH recommendations [33], the limits of detection (LOD) and limits of quantitation (LOQ) were calculated. The MW-SFL and HPLC-FD techniques yielded LOD values of 6.46 and 2.60 ng mL^−1^, respectively, and LOQ values of 19.60 and 7.81 ng mL^−1^, respectively. Table 3 provides a summary of the MW-SFL and HPLC-FD calibration and validation parameters.

#### 3.6.2. Precision and Accuracy

Using different concentration levels of the ALC sample solutions, the precision of each of the proposed MW-SFL and the HPLC-FD was computed (Table 4). The intra– and inter–assay relative standard deviations (RSD) for the MW-SFL ranged from 0.20 to 0.75 and 0.81 to 1.52%, respectively. The intra- and inter-assay RSD values for the HPLC-FD were 0.23–0.75 and 0.81–1.52 percent, respectively. Such minimal RSD values demonstrated the precision of both the MW-SFL and the HPLC-FD. The precision and accuracy of the MW-SFL and HPLC-FD were tested by employing the exact concentration levels as the recovery investigations. The recovery numbers of 96.3–100.8% (error values of −3.74–0.97%) and 96.3–102.9% (error values of −2.68–3.74%) for the MW-SFL and HPLC-FD, respectively, demonstrate the accuracy of both methodologies.

#### 3.6.3. Robustness and Ruggedness

The robustness (the effect of minor variation in the procedure on the performance of the method under investigation) of the MW-SFL and HPLC-FD methods was evaluated. It was discovered that there was no significant relationship between the amount of variation in the method variables and the analytical results produced by the method; recovery values ranged from 95.88–105.53 (±0.3–1.35%) and 95.27–105.3 (±0.22–4.48%) for the MW-SFL and HPLC-FD methods, respectively. This assures the appropriateness of both methods for the quantitation of ALC in the quality control laboratories.

Additionally, the robustness was evaluated by having two separate analysts carry out the steps of the MW-SFL and HPLC-FD methodologies on three distinct days each. Since the greatest RSD values did not exceed 5%, the results acquired from the day-to-day changes were able to be reproduced accurately.

### 3.7. Urine Analysis by the Adopted MW-SFL and HPLC-FD Methods

The recommended dose of ALC is 600 mg, taken orally twice per day for patients with ALK-positive NSCLC. Of the total dose, 98% is excreted via the feces—of which 84% is unchanged ALC—and less than 1% is found in the urine [7]. The reported ALC level in the urine was higher than the LOQ values that have been achieved by both the MW-SFL and HPLC-FD methods. Therefore, both methods could be applied to the quantitation of ALC in urine samples. The results obtained from both methods are given in Table 5. These results indicate that the mean recovery values were in the range of 91.20–100.30% (±0.25–2.5) and 95.45–101.90% (±0.96–2.25) for the MW-SFL and the HPLC methods, respectively. These results accounted for the appropriateness of the MW-SFL and HPLC-FD for routine determination of ALC in clinical laboratories.

### 3.8. A Comparison between the Proposed and Reported Methods

The results that were acquired from the presented MW-SFL and HPLC-FD methods as well as the reported liquid chromatography–mass spectrometry (LC-MS) method [19] were statistically compared, using the *t*-test and the variance ratio F-test. These tests were carried out in order to determine whether or not there was a significant difference between the three sets of results. In both the *t*-test and the variance ratio *F*-test, the computed values produced were found to be significantly lower than the tabulated ones (Table 6). According to the outcomes of this comparison, there was no statistically significant difference between the two approaches and the reported one in terms of accuracy (*t*-test) and precision (F test). Therefore, either approach may be used for the determination of the amount of ALC found in biological specimens (Table 6).

## 4. Conclusions

This study detailed the development and validation of two methods for determining the concentration of ALC in bulk and urine samples. These are the MW-SFL and HPLC-FD techniques. The MW-SFL comprised the enhancement of the native fluorescence of ALC by forming micelles with the fluorescence enhancer surfactant Cr RH 40. The method measured the fluorescence intensity using a fluorescence plate reader. For measuring the native fluorescence of ALC, HPLC-FD utilized straightforward chromatographic conditions with a short run time (<10 min) and a fluorescence detector. According to the ICH criteria for the validation of analytical processes, both methodologies were thoroughly validated giving satisfactory results. The feasibility of both techniques for ALC quantification was validated. Both the MW-SFL and HPLC-FD procedures have a high throughput because they allow the processing of a large number of samples in a short period of time, require minimal volumes of samples and organic solvents, and consequently minimize the cost of analysis and the patients’ medicine expenses. Consequently, both approaches are applicable and helpful for routine laboratory usage in determining ALC concentrations in bulk and urine samples.

## Figures and Tables

**Figure 1 medicina-59-00441-f001:**
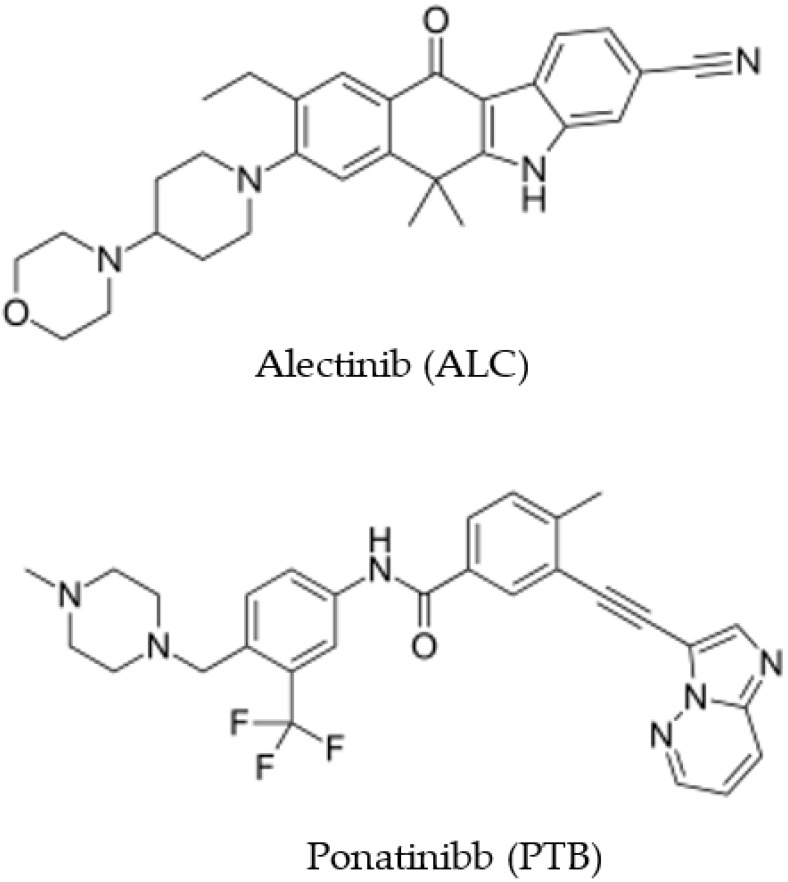
The chemical structures of alectinib (ALC) and ponatinibb (PTB).

**Figure 2 medicina-59-00441-f002:**
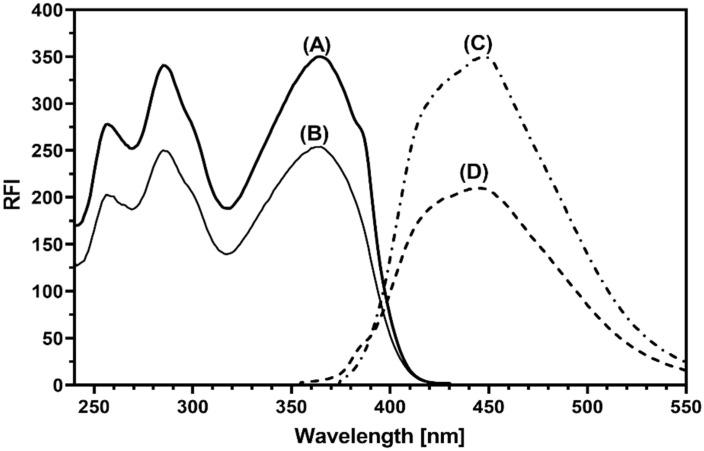
The excitation (A) and emission (C) spectra of ALC in a Cr RH 40 (1%, *w*/*v*) medium, and the excitation (B) and emission (D) spectra of ALC in aqueous media without Cr RH 40. The concentration of ALC in both cases was 250 ng mL^−1^.

**Figure 3 medicina-59-00441-f003:**
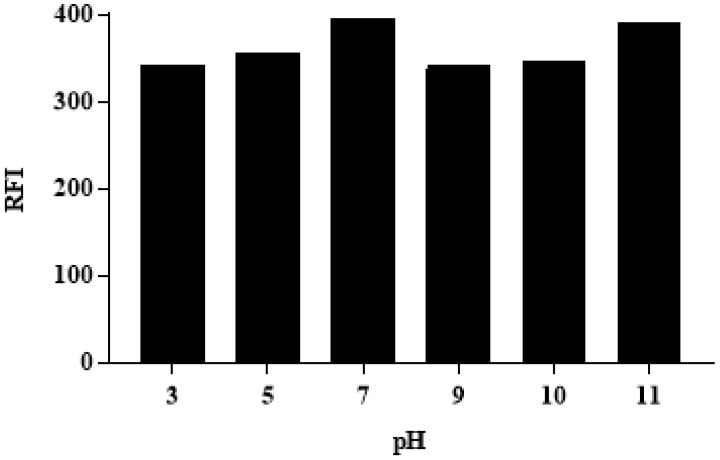
The effect of pH on the RFI of ALC (250 ng mL^−1^) in a Cr RH 40 (1%, *w*/*v*) medium.

**Figure 4 medicina-59-00441-f004:**
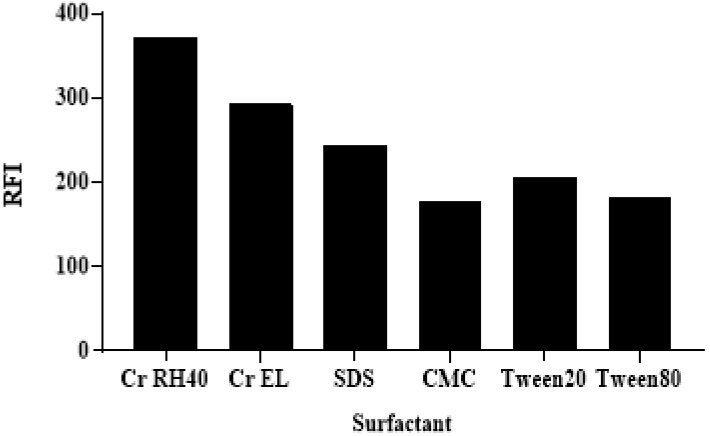
The effects of types of surfactant (1%, *w*/*v*) on the RFI of ALC (250 ng mL^−1^).

**Figure 5 medicina-59-00441-f005:**
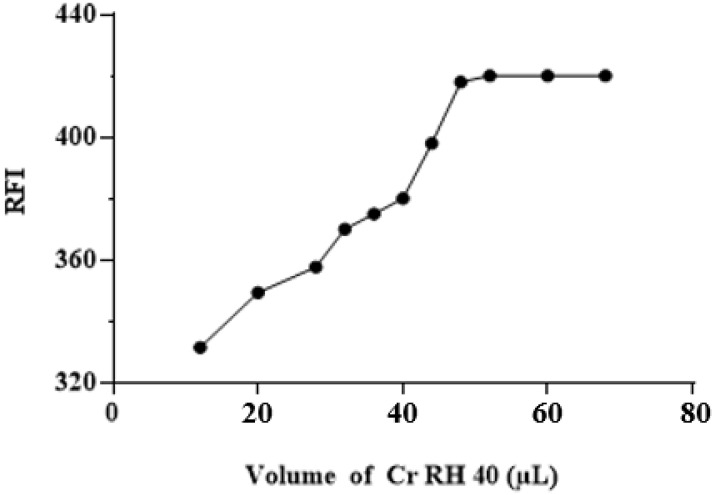
The effect of the volume of Cr RH 40 (1%, *w*/*v*) on the RFI of ALC (250 ng mL^−1^).

**Figure 6 medicina-59-00441-f006:**
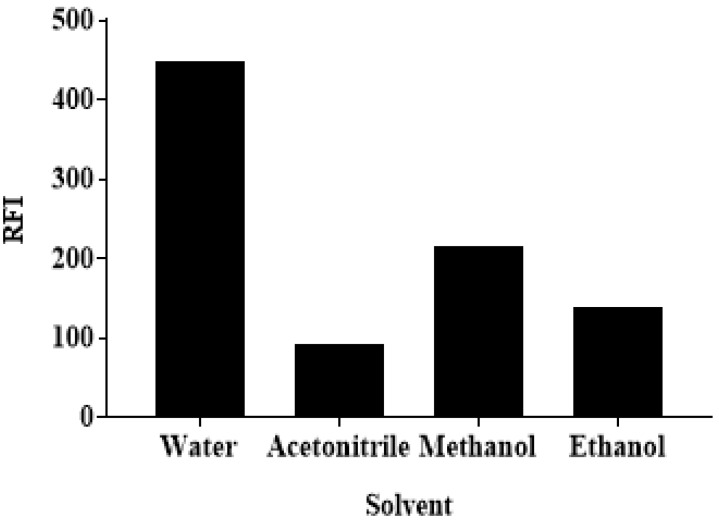
The effect of the solvent on the RFI of ALC (250 ng mL^−1^) in a Cr RH 40 (1%, *w*/*v*) medium.

**Figure 7 medicina-59-00441-f007:**
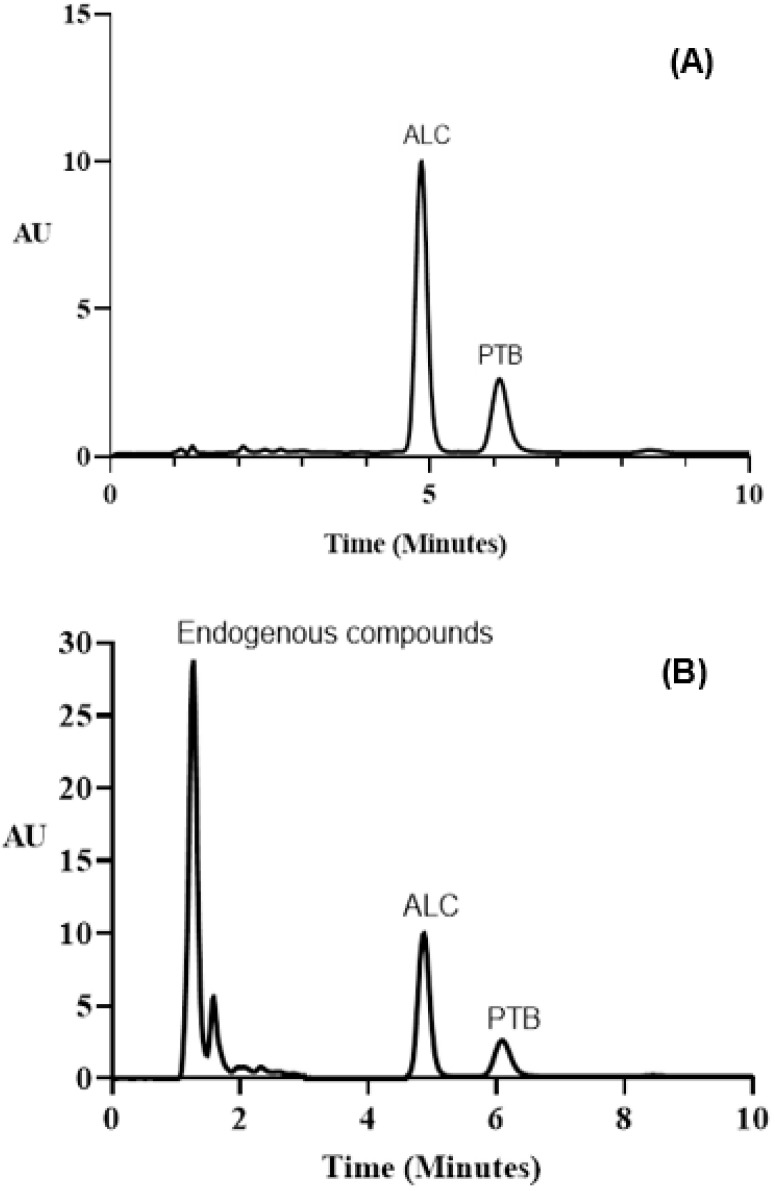
Panel (**A**), a typical chromatogram of ALC (1000 ng mL^−1^) and PTB (15.6 ng mL^−1^). Panel (**B**), a typical chromatogram of a urine sample spiked with ALC (1000 ng mL^−1^) and PTB (15.6 ng mL^−1^). AU (y-axis) stands for arbitrary units.

**Figure 8 medicina-59-00441-f008:**
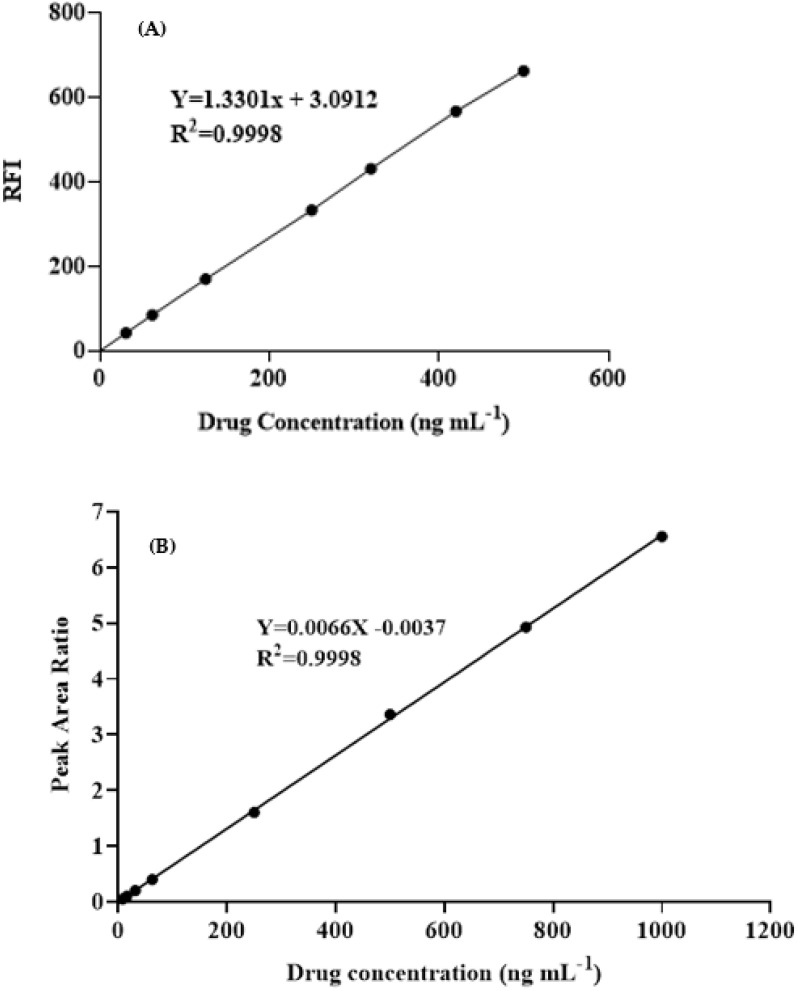
Standard calibration curves for the determination of ALC by the proposed MW-SFL (**A**) and HPLC-FD (**B**) methods.

**Table 1 medicina-59-00441-t001:** The optimization of experimental conditions for the MW-SFL and HPLC-FD methods for determination of ALC.

Method/Condition	Studied Range	Optimum Value
MW-SFL		
Surfactants (1%, *w*/*v*)	Different ^a^	Cr RH40
Volume of Cr RH 40 (µL)	15–70	60
Solvent	Different ^b^	Water
pH	3–11	7
Excitation wavelength (λ_ex_, nm)	230–440	365
Emission wavelength (λ_em_, nm)	350–550	450
HPLC-FD		
Organic solvent in a mobile phase	Acetonitrile, methanol	Methanol
Phosphate buffer:methanol ratio (%)	60–90	80
pH of buffer solution	4–7	7
Flaw rate (mL min^–1^)	0.5–2 mL min^–1^	2
Internal standard	Different ^c^	Pontatinib
Excitation wavelength (nm)	230–440	365
Emission wavelength (nm)	350–550	450

^a^ The surfactants used were Cremophor RH40 (Cr RH 40), Cremophor EL, sodium dodecyl sulfate (SDS), carboxymethyl cellulose (CMC), Tween 20, and Tween 80. ^b^ The solvents tested were water, acetonitrile, methanol, and ethanol. ^c^ The internal standards tested were ponatinib, erlotinib, dasatinib, and vandetanib.

**Table 2 medicina-59-00441-t002:** Chromatographic parameters for ALC by the proposed HPLC-FD method.

Parameter	Value	Recommended Value
Retention time (min)	4.86	
Capacity factor (*K`*)	3.5	1–10
Separation factor (α)	1.33	>1
Resolution factor (Rs)	2.97	≥1.5
Peak asymmetry factor	1.10	≥1
Number of theoretical plates (*N*) per meter	1794	Varied

**Table 3 medicina-59-00441-t003:** Regression and statistical parameters for the determination of ALC by the proposed MW-SFL and HPLC-FD methods.

Parameter	Value ^a^	
	MW-SFL	HPLC-FD
Range (ng mL^−1^)	30–500	5–1000
Intercept	3.09	0.0037
Slope	1.33	0.0066
Determination coefficient (r^2^)	0.9998	0.9998
LOD ^b^ (ng mL^−1^)	6.46	2.60
LOQ ^c^ (ng mL^−1^)	19.60	7.81

^a^ Average of 3 measurements. ^b^ Limit of detection. ^c^ Limit of quantification.

**Table 4 medicina-59-00441-t004:** The precision and accuracy of both the MW-SFL and the HPLC-FD methods for determination of ALC.

ALC Concentration (ng mL^−1^)	Intra-Day ^a^		Inter-Day ^b^	
	Recovery (% ± RSD)	Error (%)	Recovery (% ± RSD)	Error (%)
MW-SFL				
125	99.8 ± 0.75	−0.19	100.2 ± 1.52	−0.21
250	98.5 ± 0.23	−1.49	99.0 ± 0.81	0.97
420	96.3 ± 0.70	−3.74	100.8 ± 0.88	0.81
HPLC-FD				
250	96.3 ± 0.58	3.70	97.3 ± 1.09	2.7
500	101.8 ± 0.70	−1.80	98.4 ± 2.32	1.6
750	102.7 ± 0.23	−2.68	102.9 ± 0.34	2.8

^a^ Average of 3 measurements. ^b^ Average of 6 measurements.

**Table 5 medicina-59-00441-t005:** The analysis of ALC in the spiked human urine samples by the proposed MW-SFL and HPLC-FD methods.

Spiked Concentration (ng mL^−1^)	Measured Concentration (ng mL^−1^)	Recovery (% ± SD) ^a^	Error (%)
MW-SFL			
60	54.54	91.20 ± 0.25	8.79
120	117.90	98.25 ± 1.55	3.09
240	231.87	96.61 ± 2.50	3.38
480	481.46	100.30 ± 1.02	−0.30
HPLC-FD			
120	116.94	95.45 ± 2.25	4.55
240	244.56	101.90 ± 1.35	−1.90
480	478.46	99.68 ± 0.96	0.32

^a^ Values are average of 3 measurements.

**Table 6 medicina-59-00441-t006:** The statistical analysis of the results obtained via the proposed MW-SFL and HPLC-FD methods and the reported LC-MS method for the analysis of ALC in spiked human urine sample.

Value	MW-SFL	LC-MS	HPLC-FD	LC-MS
Mean (%)	99.40	98.84	97.87	99.40
RSD (%)	2.28	3.87	2.14	3.87
*N*	7	5	7	5
*t*-test	1.15 (2.77) ^a^		0.44 (2.77) ^a^	
*F*-test	2.89 (4.53) ^b^		3.37 (4.53) ^b^	

^a^ Tabulated value of *t*-test at a confidence level of 95%. ^b^ Tabulated value of F-test at a confidence level of 95%.

## Data Availability

All data are in the article.

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
