# Peer review of "Development of Novel Microwell-Based Spectrofluorimetry and High-Performance Liquid Chromatography with Fluorescence Detection Methods and High Throughput for Quantitation of Alectinib in Bulk Powder and Urine Samples"

_medicina, 2023, doi:10.3390/medicina59030441_

Round 1
Reviewer 1 Report
Dear Authors,
I found this manuscript "Development of novel microwell-based spectrofluorimetry and high-performance liquid chromatography with fluorescence detection methods with high throughput for quantitation of alectinib in bulk powder and urine samples" interesting, but I have several comments:
- I suggest naming these materials (Cremophor® RH 40, Cremophor® EL, Tween® 20, and Tween® 80) not only by brand names but by chemical names. Lines 331–332, 335
- What was the freezing temperature? It should be mentioned in the text. Line 341
- You write "1 mL of human urine samples were spiked with 20 L of ALC standard solution containing different amounts of ALC, ...". Was the volume of the ALC standard solution really 20 L, or maybe 20 mL or 20 µL? Line 350
- What device did you use to fix the duration in seconds (30 s, 10 s)? Lines 351–353
- There are given only 1% of Cr RH 40 solution in the method (3.5.1. MW-SFL). Were aqueous solutions of other surfactants used in this study? Line 363
- What does RFI mean? Line 364
- You write "The volume of the sample injection was 10 L". Maybe 10 µL? Line 372
- You write "The spectra of ALC showed that ALC has three excitation wavelengths (λex) at 360, 275 and 365 nm and". Maybe 260, 275, and 365 nm? Line 101
- You write "2.3. Optimization of MW-SFL method variables and 2.5.2. Optimization of chromatographic conditions". What experimental design (DoE) did you use to optimize the method variables or chromatographic conditions? According to your description of these methods, you just selected the most suitable conditions of the variables but did not optimize them. Lines 112–158, 184–234
- You write "... the maximum RFI value was obtained at pH 7 (Fig. 3)...". However, according to Figure 3, the high RFI value also was obtained at pH 11. Please comment. Line 116
- You write "The obtained results (Table 2) were found in accordance with and demonstrating good performance efficiency". Please supplement Table 2 with recommended values according to International Council for Harmonization (ICH) guidelines. Lines 224–228
- You write "HPLC-FD and MW-SFL both achieved LOQ values of 19.60 and 7.81 MW-SFL ng mL-1, respectively". However, according to Table 3, it should be the opposite (MW-SFL 19.60 ng mL-1 and HPLC-FD 7.81 ng mL-1). Lines 251–252
- You write "The intra- and inter-assay relative standard deviations (RSD) for MW-SFL ranged from 0.20 to 0.75 and 0.81 to 1.52%, respectively". However, according to Table 4, it should be 0.23 to 0.75 %. Line 257
- You write "Such minimal RSD values demonstrated the accuracy ...". However, instead of accuracy, there should be precision. Line 259
- You write "The recovery ... demonstrate the precision of both methodologies". However, instead of precision, there should be accuracy. Lines 261–264
- Should be error values of –2.68–3.70% (according to Table 4). Line 262
- Instead of 86.90–100.30% (± 0.25–2.5), there should be 91.20–100.30% (± 0.25–2.5) (according to Table 5). Line 287
Author Response
Please see the attchment

Reviewer 2 Report
· Justification is required? precision and accuracy of photoluminescence methods are usually poorer than spectrophotometric procedures by a factor of perhaps 2 to 5. The precision of photoluminescence methods is often limited by source flicker noise and drift. The accuracy is often limited by concomitants, or particles, in the sample that causes additional fluorescence and scattering or that quench the analyte fluorescence.
· Abbreviations may be expanded e.g. RFI
· 2.7 the word “clinal” should be corrected

Round 2
Reviewer 1 Report
Dear Authors,
I disagree with some of your answers and corrections:
- You write "2.3. Optimization of MW-SFL method variables and 2.5.2. Optimization of chromatographic conditions". What experimental design (DoE) did you use to optimize the method variables or chromatographic conditions? According to your description of these methods, you just selected the most suitable conditions of the variables but did not optimize them. Lines 112–158, 184–234
We did not use DoE; however, the optimization was carried out experimentally. We have done all the optizations and selected the most appropriate condition for establishing the final procedures of both MW-SFA and HPLC-FD.
Comment: if you didn't use the experimental design (DoE), then you just chose the best conditions from the studied ones. Can you guarantee that there are no better conditions? I suggest using selection instead of optimization.
- You write "Both the MW-SFL and HPLC-FD techniques yielded LOD values of 2.6 and 6.46 ng mL-1". However, according to Table 3, it should be the opposite (MW-SFL 6.46 ng mL-1 and HPLC-FD 2.60 ng mL-1). Lines 256–257.
- Should be error values of –2.68–3.70% (according to Table 4). Line 262
This error was corrected (line 257).
Comment: 2.68 must have a minus sign. Line 270
